# Benefits of a Single Dose of Betamethasone in Imminent Preterm Labour

**DOI:** 10.3390/jcm11010020

**Published:** 2021-12-21

**Authors:** Natalia Saldaña-García, María Gracia Espinosa-Fernández, Celia Gómez-Robles, Antonio Javier Postigo-Jiménez, Nicholas Bello, Francisca Rius-Díaz, Tomás Sánchez-Tamayo

**Affiliations:** 1Department of Neonatology, Regional University Hospital of Malaga, 29010 Malaga, Spain; mgespinosaf@gmail.com (M.G.E.-F.); celiagr77@hotmail.com (C.G.-R.); ajpostigojimenez@gmail.com (A.J.P.-J.); nicholasbello90@gmail.com (N.B.); tomas.sanchez.tamayo@gmail.com (T.S.-T.); 2School of Medicine, Malaga University, 29071 Malaga, Spain; 3Department of Preventive Medicine and Public Health, Biostatistics, School of Medicine, Malaga University, 29071 Malaga, Spain; rius@uma.es; 4Pediatrics Division, Malaga University, 29071 Malaga, Spain

**Keywords:** antenatal corticosteroids, betamethasone, preterm infant, mortality, respiratory distress syndrome

## Abstract

Background: A complete course of prenatal corticosteroids reduces the possibility of morbimortality and neonatal respiratory distress syndrome (RDS). Occasionally, it is not possible to initiate or complete the maturation regimen, and the preterm neonate is born in a non-tertiary hospital. This study aimed to assess the effects of a single dose of betamethasone within 3 h before delivery on serious outcomes (mortality and serious sequelae) and RDS in preterm neonates born in tertiary vs. non-tertiary hospitals. Materials and methods: Preterm neonates who were <35 weeks and ≤1500 g, treated during a period of five years in a level IIIC NICU, were included in this retrospective cohort study. Participants were divided into groups as follows: NM, non-matured; PM, partial maturation (one dose of betamethasone up to 3 h antepartum). They were further divided based on their place of birth (NICU-IIIC vs. non-tertiary hospitals). The morbimortality rates and the severity of neonatal RDS were evaluated. Results: A total of 76 preterm neonates were included. A decrease in serious outcomes was found in the PM group in comparison to the NM group (OR = 0.2; 95%CI (0.07–0.9)), as well as reduced need for mechanical ventilation (54% vs. 68%). The mean time between maternal admission and birth was similar in both cohorts. The mean time from the administration of betamethasone to delivery was 1 h in the PM cohort. With regard to births in NICU-IIIC, the PM group performed better in terms of serious outcomes (32% vs. 45%) and the duration of mechanical ventilation (117.75 vs. 132.18 h) compared to the NM group. In neonates born in non-tertiary hospitals with PM in comparison to the NM group, a trend towards a reduced serious outcome (28.5% vs. 62.2%) and a decreased need for mechanical ventilation (OR = 0.09; 95%CI (0.01–0.8)) and maximum FiO_2_ (*p* = 0.01) was observed. Conclusions: A single dose of betamethasone up to 3 h antepartum may reduce the rate of serious outcomes and the severity of neonatal RDS, especially in non-tertiary hospitals.

## 1. Introduction

The benefits of antenatal corticosteroid (ACS) administration in terms of reduced mortality and neonatal respiratory distress syndrome (RDS) rates in the preterm population are well documented in the scientific literature. Exposure to antenatal corticosteroid has also been associated with a decrease in the occurrence of intraventricular haemorrhage (IVH) [1].

These results have been verified for the full corticosteroid course, i.e., two doses of betamethasone, 12 mg each, 24 h apart, or four doses of dexamethasone, 6 mg each, 12 h apart, when preterm birth occurs within one to seven days of complete maturation [1,2,3].

With regard to precipitous preterm deliveries or situations that have imminent risk to the mother or foetus, at least one dose of antenatal corticosteroid is recommended, even if the completion of the course is not possible. However, on some occasions when a rapid birth is anticipated, the delivery may be completed without any prenatal corticosteroid dose [2,4].

The minimum time interval between the administration of antenatal corticosteroids and the appearance of significant beneficial effects for the preterm infant has not been determined. A mathematical estimate has recently been published in the EPICE study, which concludes that a single dose of corticosteroid administered 3 h before birth could lead to a decrease in mortality rate of up to 26% in the preterm population [5]. 

Situations regarding imminent delivery, either due to maternal, foetal, or membrane pathology, require rapid action by the obstetric team and may occur in both tertiary and secondary centres. Given the urgency of these events, on some occasions, it is not possible to transfer the pregnant woman from the regional hospital to the referral hospital, and the preterm is born in level I/II neonatal units, with subsequent transfer of the neonate, following stabilisation, to the referral level III unit for treatment [6].

The aim of this study was to assess the effects of a single dose of betamethasone administered within 3 h prior to birth with regard to serious outcomes and neonatal respiratory distress syndrome (RDS) in preterm infants born in tertiary vs. secondary hospitals. 

## 2. Materials and Methods

A retrospective cohort study was conducted from 1 January 2015 to 31 December 2020, including preterm infants under 35 weeks of gestational age (GA), weighing ≤ 1500 g, admitted in a Neonatal Intensive Care Unit (NICU) IIIC level (Regional University Hospital of Málaga). Neonates with major malformations, genetic syndromes, and a prenatal diagnosis of intrauterine growth retardation were excluded from the study. Informed consent was obtained from parents upon admission to the NICU. 

The study population was divided based on the course of lung maturation administered: non-matured (NM), which included preterm infants who received no antenatal corticosteroid dose, and partial maturation (PM), which included preterm infants who received a single dose of betamethasone (12 mg) within 3 h before birth. 

The following variables were recorded as demographic and clinical characteristics: maternal age, primiparity, single/multiple gestations, the presence of gestational diabetes, hypertensive stages including pre-eclampsia, eclampsia and HELLP syndrome [7], chorioamnionitis [8,9] and third trimester haemorrhage (placental abruption, placenta praevia, uterine rupture, or vasa praevia). The following data were collected: place of birth, caesarean section, gestational age (GA) in weeks, anthropometric measurements at birth, gender, Apgar test score < 5 at 5 min of life and need for intubation at birth.

The main objective was to determine “serious outcomes”, defined as death or survival with serious sequelae, which included the following: intraventricular haemorrhage grade III–IV (IVH), according to Papile classification [10]; periventricular leukomalacia (PVL) in either the cystic or non-cystic form, defined as changes in the signal intensity or echogenicity of the periventricular white matter, detected by ultrasound or MRI [11]; retinopathy of prematurity (ROP) requiring treatment [12]; necrotising enterocolitis (NEC) stage ≥ 2 according to the Bell classification [13]; moderate (the need for supplemental O_2_ for ≥28 days and FiO_2_ < 30% at 36 weeks postmenstrual age or discharge) or severe (the need for supplemental O_2_ for ≥28 days greater than 30% and/or continuous positive pressure or mechanical ventilation at 36 weeks postmenstrual age or discharge) bronchopulmonary dysplasia (BPD) [14].

As secondary outcomes, the severity of RDS was determined by establishing the need for mechanical ventilation (MV), time on VM, maximum FiO_2_ during admission, and the need for surfactant administration. Other variables analysed included hypotension during the first week of life requiring treatment (volume expansion or inotropic administration), persistent ductus arteriosus with haemodynamic repercussion requiring medical or surgical treatment, and the presence of sepsis (defined as clinical signs of sepsis and/or suggestive laboratory test results with a confirmatory blood culture), either early (in the first 72 h of life) or late (after the first 72 h of life) [15].

After the baseline characteristics of the study population were assessed, the population was further divided based on the place of birth (neonates born in the tertiary hospital with NICU-IIIC and neonates born in secondary hospitals and subsequently transferred to the NICU-IIIC).

Finally, in order to determine whether there would have at least been sufficient time to administer a dose in the non-matured (NM) group, the time elapsed from the admission of the pregnant woman to the emergency department to the birth of the preterm infant, in minutes, was analysed with regard to the PM and NM cohorts. This analysis was completed for the PM population using the time interval from maternal admission to prenatal corticosteroid administration and from prenatal corticosteroid administration to birth.

Contingency tables and the chi-squared test were used for the comparison of qualitative variables. In the 2 × 2 tables with a low number of observations (*n* < 5), Fisher’s exact test and the odds ratios were calculated. To carry out pairwise comparisons of quantitative variables the Student’s *t*-test was used. In all cases, a statistically significant difference was declared when the level for statistical significance was found to be below 5% (*p* < 0.05). For statistically significant results, a multivariate model was adjusted for the following confounding factors: GA, the presence of chorioamnionitis, hypertensive stages, and Apgar test score < 5 at 5 min of life. The statistical program SPSS 25.0 (IBM Corp., Armonk, NY, USA) was used.

This study was approved by the Provincial Ethics Committee of Málaga and by the Medical Management of the Regional University Hospital of Málaga.

## 3. Results

A total of 76 patients were studied, who were divided into the NM cohort (*n* = 41) and PM cohort (*n* = 35). The characteristics of the two cohorts are summarised in Table 1. The NM cohort had a higher percentage of secondary hospital births. The other variables, including GA, were distributed among the two cohorts without significant differences.

A comparison analysis between the NM versus PM cohort is shown in Table 2. The rate of serious outcomes was found to be lower among those who received a single dose of betamethasone within 3 h prior to birth (PM cohort) than in those who received none, both in the unadjusted and in the multivariable model (OR 0.2 95%CI (0.07–0.9)). Likewise, preterm infants who received a pre-birth dose showed lower percentage rates of periventricular leukomalacia (PVL) and treated ROP. Trends that favoured the PM cohort were found in relation to other outcome variables, which included: less need for MV, shorter MV times, lower maximum FiO_2_ during admission, less hypotension requiring treatment during the first week of life, and reduced rates of moderate/severe BPD.

Data concerning the time elapsed (in minutes) from the admission of the pregnant woman to preterm birth are shown in Figure 1. Surprisingly, somewhat longer periods of time elapsed with regard to the NM cohort, although the difference did not reach statistical significance (NM vs. PM, mean ± standard deviation: 150.7 ± 118.9 vs. 126.1 ± 57.5, respectively).

For the PM cohort, the time intervals from the admission of the pregnant woman to antenatal betamethasone administration and from corticosteroid administration to preterm birth were found to be 35.4 ± 58.2 and 61.8 ± 44.4 min, respectively.

An analysis of the NM and PM cohorts of preterm infants born at the tertiary hospital was performed (Table 3). In the analysis of NM versus PM, a trend towards lower serious outcome rates in the group receiving a single dose of antenatal corticosteroid was seen. Better results were also observed in terms of the rates of PVL, severe IVH, moderate/severe BPD, arterial hypotension in the first week of life, and the need for surfactant therapy, as well as with regard to MV time.

The analysis of the effects of partial maturation in the secondary hospital population is shown in Table 4. The decrease in the number of cases had an impact on the statistical power of this comparison, although the trends towards better results in relation to the PM cohort with regard to the rates of serious outcomes and different morbidities (severe IVH, PVL, ROP, and arterial hypotension during the first week of life) were maintained. There was a significant decrease in the need for MV that favoured the PM cohort, which was maintained following adjustment for confounding factors (OR 0.09 95%CI (0.01–0.8)). The maximum FiO_2_ during admission was significantly lower in the PM group, *p* = 0.01; 95%CI (−45.19, −5.63). Time on VM was also significantly lower in the PM group, but when it was adjusted for confounding factors, significance was lost (*p* = 0.19).

A further analysis was performed to assess the effect of birth in secondary hospitals and the subsequent transfer of the preterm infant with regard to the non-matured and partially matured populations (Table 5 and Table 6, respectively). The analysis of the NM group revealed worse results for the group born in the secondary hospital, with a significantly higher rate of the need for MV in both the unadjusted and confounder-adjusted models (OR 9.05 95%CI (1.62–50.62)). Comparison between the PM groups based on place of birth showed similar percentage rates of serious outcome and different morbidities. The tertiary hospital PM cohort had a longer MV time; however, when it was adjusted for confounding factors, this significance disappeared (*p* = 0.46; OR 0.38 95%CI (−126.6–9)).

## 4. Discussion

The time interval between the administration of antenatal corticosteroid and preterm birth is of utmost importance, as the beneficial effects of corticosteroids on the rates of serious outcomes and respiratory distress depend on this interval [3,16,17,18].

There are clinical situations where prenatal corticosteroid therapy is not initiated because birth is considered imminent, or the full course cannot be completed. Additionally, in these high-risk situations, the cascade of events prevents the transfer of the pregnant woman if she is in a non-tertiary hospital, and the premature birth takes place in a level I/II neonatal unit [6,19].

Our results show a 22.3% decrease in serious outcomes in favour of the group that received a single dose of betamethasone immediately before birth, which is in line with the results found in the literature [5,16]. In the EPICE study published by Norman et al. (2017), a mathematical estimate was made based on the results obtained from a cohort of 4594 preterm infants, concluding that a single dose of corticosteroid administered within 3 h before birth could reduce the mortality rate associated with non-matured neonates up to 26% [5]. In our study, we found no significant differences in mortality rates, probably due to insufficient sample size (289 infants in each arm would have been needed to obtain sufficient power to detect a 20% decrease in mortality). The minimum time interval between prenatal corticosteroid administration and preterm birth to obtain the benefits of lung maturation has not been determined. In our study, with a mean of 61.8 ± 44.4 min between the administration of a single dose of corticosteroid and preterm birth and up to a maximum of 3 h, better results were found for the PM cohort.

In animal models, betamethasone levels that are effective for lung maturation (1–4 ng/mL) have been detected in the umbilical cord at the time of prenatal administration. [20] Based on these models, it was determined that in pregnant women > 28 weeks, the intramuscular administration of 11.4 mg of betamethasone produces levels > 1 ng/mL in the umbilical cord from the first hour of administration and are maintained for up to 1.4 days. Levels below 1 ng/mL increase the risk of surfactant therapy [21]. The placental transfer of betamethasone has been compared in single gestational, twin, and obese mothers, with no differences and levels > 1 ng/mL after the first dose of betamethasone [22]. These findings may support the biological plausibility of our results, although further studies are needed to analyse betamethasone levels in the umbilical cord after administration close to delivery.

In some situations where delivery is imminent, prenatal corticosteroids were not administered because it was considered that there would not be enough time for them to have an effect on the neonate. In our study, the timeline from the moment a pregnant woman was admitted to the emergency department to the birth of a preterm infant was recorded. This time interval was similar for non-matured and partially matured neonates (with a mean of 150 min for the NM group vs. 126 min for the PM group). Our results indicate that some of the preterm infants in the non-matured cohort could have benefited from the effects of a single dose of antenatal betamethasone.

Regarding the results found in preterm infants born in non-tertiary hospitals, the beneficial effect of antenatal betamethasone was evident. The PM cohort was found to exhibit a trend towards lower serious outcome rates. In addition, there was less need for MV and lower maximum FiO_2_ in the group that received prenatal corticosteroids, although the sample of preterm infants for the PM cohort was smaller. In our study, a pre-delivery dose of betamethasone was shown to improve the outcome of preterm infants who had to undergo inter-hospital transfer to continue their treatment [19].

Mixed analysis between NM and PM cohorts born in secondary hospitals and the tertiary hospital reinforces the recommendations that favour preterm birth to take place in specialised units [6,19].

The strengths of this study include the homogeneity of the cohorts in terms of gestational age, birth weight, and perinatal history. Subsequent analysis controlling for confounding factors consolidated the results. The biological plausibility and the results according to the literature, support this research.

Several limitations of our study have to be pointed out. First, the retrospective approach of this research prevents from obtaining definitive conclusions about single-dose maturation, although it poses an important hipothesys to be tested in a larger prospective clinical trial. Second, the small sample size of the study population probably had an influence on not reaching significant differences in some of the variables analysed. Despite this, trends towards better results were observed in the group that received at least one dose. Finally, long-term outcomes have not been addressed in our study.

Given the international recommendation to initiate maturation with prenatal corticosteroids in situations of preterm birth, there are few cases in which maturation is not administered or in which it is administered just before delivery [23]. Further studies are necessary to confirm our results and determine the minimum time it takes for benefits from prenatal corticosteroid to be obtained by premature infants.

## 5. Conclusions

In conclusion, the administration of a single dose of betamethasone within 3 h prior to preterm birth may decrease the rates of serious outcomes and other morbidities in neonates. In particular, beneficial effects were observed when delivery takes place in non-tertiary hospitals. Given the available scientific evidence and the results of this study, we consider that urgent administration of antenatal betamethasone may be a safe and effective strategy in cases of imminent preterm birth.

## Figures and Tables

**Figure 1 jcm-11-00020-f001:**
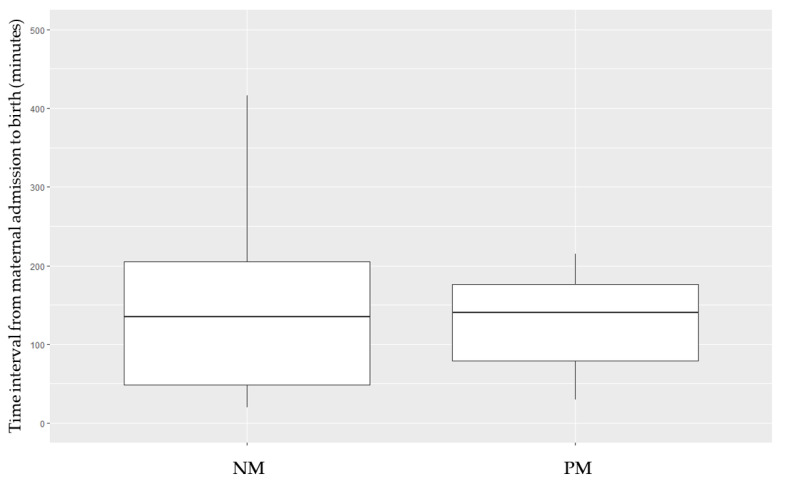
Time interval (in minutes) from the admission of the pregnant woman to the birth of the preterm neonate, depending on the maturation pattern received. NM, non-matured. PM, partial maturation.

**Table 1 jcm-11-00020-t001:** Characteristics of the cohorts included in the study according to the maturation pattern.

	NM (*n* = 41)	PM (*n* = 35)	Significance (*p*) NM vs. PM *
Maternal age (years)	31.39 ± 7.06	30.55 ± 6.69	0.60
Primiparity	13 (38.2%)	11 (32.4%)	0.61
Multiple gestation	11(26.8%)	7 (20%)	0.48
Gestational diabetes	2 (6.9%)	0	0.49
Hypertensive states	3 (7.7%)	1 (2.9%)	0.61
Chorioamnionitis	12 (30%)	9 (25.7%)	0.68
Third trimester haemorrhage	10 (25%)	6 (17.15%)	0.31
Secondary hospital birth	19 (46.3%)	7 (20%)	0.01
Caesarean section	26 (63.4%)	2 (65.7%)	0.92
Gestational age (weeks)	28.03 ± 2.61	27.84 ± 2.47	0.75
Birth weight (grams)	1040.98 ± 280.6	1025.9 ± 291.7	0.82
Height at birth (cm)	37.05 ± 3.1	35.7 ± 3.5	0.16
Head circumference at birth (cm)	26.65 ± 2.2	25.3 ± 2.8	0.08
Gender (female)	18 (43.9%)	20 (57.14%)	0.25
Apgar < 5 at 5 min	5 (14.3%)	2 (5.7%)	0.42
Intubation at birth	28 (68.3%)	20 (57.1%)	0.35

Qualitative variables are expressed as *n* (%); quantitative variables as mean ± standard deviation. * Fisher’s exact test/chi-squared test for qualitative variables; Student’s *t*-test for quantitative variables. NM, not matured; PM, partial maturation.

**Table 2 jcm-11-00020-t002:** Neonatal outcomes of cohort comparison according to maturation pattern.

	NM (*n* = 41)	PM (*n* = 35)	Significance (*p*) *	OR(95%CI) Unadjusted	OR(95%CI) Adjusted ^‡^
Exitus	6 (14.6%)	7 (20%)	0.53	1.4 (0.44–4.83)	
Serious outcome	22 (53.7%)	11 (31.4%)	0.05	0.3 (0.15–1)	0.2 (0.07–0.9)
Severe IVH	7 (17.1%)	5 (14.3%)	0.74	0.8 (0.2–2.8)	
PVL	5 (12.2%)	0	0.05	0.8 (0.7–0.9)	NA
NEC ≥ 2nd degree	1 (2.6%)	0	1	0.9 (0.92–1)	NA
Treated ROP	5 (12.2%)	0	0.05	0.8 (0.7–0.9)	NA
Arterial hypotension	17 (42.5%)	11 (31.4%)	0.32	0.6 (0.2–1.6)	
Treated PDA	14 (34.1%)	16 (45.7%)	0.42	1.6 (0.6–4.1)	
Moderate/severe BPD	10 (28.6%)	3 (10.7%)	0.11	0.3 (0.07–1.2)	
Surfactant requirement	33 (80.5%)	28 (80%)	0.95	0.9 (0.31–3)	
MV	28 (68.3%)	19 (54.4%)	0.21	0.5 (0.2–1.4)	
Early sepsis	2 (5.1%)	1 (2.9%)	1	0.5 (0.04–6.2)	
Late sepsis	14 (35.9%)	10 (28.6%)	0.62	0.7 (0.2–1.9)	
			Significance (*p*) ^†^		
MV time (hours)	159.42 ± 176.51	98.84 ± 81.17	0.12		
Maximum FiO_2_	50.12 ± 26.37	42.32 ± 24.77	0.19		

Variables are expressed as *n* (%) and mean ± standard deviation for qualitative and quantitative variables, respectively. * Fisher’s exact test/chi-squared test; ^†^ Student’s *t*-test. ^‡^ Results of multivariate analysis are shown for significant variables in the unadjusted model. NM, not matured; PM, partial maturation; IVH, intraventricular haemorrhage; PVL, periventricular leukomalacia; NEC, necrotising enterocolitis; ROP, retinopathy of prematurity; PDA, patent ductus arteriosus; BPD, bronchopulmonary dysplasia; MV, mechanical ventilation. NA, not applicable, as one of the cohorts analysed had a result of “0” for the variable.

**Table 3 jcm-11-00020-t003:** Comparison of the tertiary hospital birth population based on the maturation pattern received.

	NM (*n* = 22)	PM (*n* = 28)	Significance (*p*) *	OR (95%CI)
Exitus	5 (22.7%)	6 (21.4%)	0.91	0.9 (0.24–3.55)
Serious outcome	10 (45.5%)	9 (32.1%)	0.33	0.5 (0.18–1.80)
Severe IVH	3 (13.6%)	4 (14.2%)	1	1.2(0.21–5.29)
PVL	2 (9.1%)	0	1	0.9 (0.79–1.03)
NEC ≥ 2nd degree	1 (5%)	0	0.41	0.9 (0.85–1.05)
Treated ROP	2 (9.1%)	0	0.18	0.9 (0.79–1.03)
Arterial hypotension	8 (38.1%)	9 (32.1%)	0.76	0.7 (0.23–2.51)
Treated PDA	6 (27.6%)	13 (46.3%)	0.16	2.3 (0.69–7.64)
Moderate/severe BPD	3 (17.6%)	3 (13.6%)	1	0.7 (0.12–4.21)
Surfactant requirement	18 (81.8%)	22 (78.5%)	0.77	0.8 (0.19–3.33)
MV	11 (50%)	16 (57.1%)	0.61	1.3 (0.43–4.09)
Early sepsis	1 (4.5%)	1 (3.5%)	1	0.7(0.04–11.91)
Late sepsis	5 (22.7%)	9 (32.1%)	0.75	1.4 (00.39–5.14)
			(*p*) ^†^	
MV time (hours)	132.18 ± 186.91	113.75 ± 79.82	0.76	
Maximum FiO_2_	45.80 ± 27.96	45.92 ± 26.18	0.98	

Results are expressed as *n* (%) and mean ± standard deviation for qualitative and quantitative variables, respectively. * Fisher’s exact test/chi-squared test; ^†^ Student’s *t*-test. NM, non-matured; PM, partial maturation; CM, complete maturation; IVH, intraventricular haemorrhage; PVL, periventricular leukomalacia; NEC, necrotising enterocolitis; ROP, retinopathy of prematurity; PDA, patent ductus arteriosus; BPD, bronchopulmonary dysplasia; MV, mechanical ventilation.

**Table 4 jcm-11-00020-t004:** Comparison of the secondary hospital birth population based on the maturation pattern received.

	NM (*n* = 19)	PM (*n* = 7)	Significance (*p*) *	OR (95%CI) Unadjusted	OR (95%CI) Adjusted ^‡^
Exitus	1 (5.3%)	1 (14.3%)	0.47	3 (0.16–55.72)	
Serious outcome	12 (62.2%)	2 (28.5%)	0.19	0.2 (0.03–1.5)	
Severe IVH	4 (21.1%)	1 (14.2%)	1	0.6 (0.05–6.8)	
PVL	3 (15.8%)	0	0.54	0.8 (0.6–1.02)	
NEC ≥ 2nd degree	0	0	-	-	-
Treated ROP	3 (15.8%)	0	0.54	0.8 (0.6–1.02)	
Arterial hypotension	9 (47.4%)	2 (28.5%)	0.65	0.4 (0.06–2.88)	
Treated PDA	8 (42.1%)	3 (42.8%)	1	1.03 (0.17–5.9)	
Moderate/severe BPD	7 (38.9%)	0	0.13	0.2 (0.02–2.09)	
Surfactant requirement	15 (78.9%)	6 (85.7%)	0.69	1.6 (0.14–17.1)	
MV	17 (89.5%)	3 (42.8%)	0.02	0.08 (0.01–0.71)	0.09 (0.01–0.8)
Early sepsis	1 (5.3%)	0	1	0.94 (0.85–1.05)	
Late sepsis	9 (47.4%)	1 (14.3%)	0.19	0.18 (0.02–1.84)	
			Significance (*p*) ^†^		Adjusted significance (*p*) ^‡^
MV time (hours)	177.05 ± 172.92	19.33 ± 16.04	0.002		0.19
Maximum FiO_2_	55.16 ± 24.17	28.42 ± 11.04	0.01		0.01

Variables are expressed as *n* (%) and mean ± standard deviation for qualitative and quantitative variables, respectively. * Fisher’s exact test/chi-squared test; ^†^ Student’s *t*-test. ^‡^ Results of multivariate analysis are shown for significant variables in the unadjusted model. NM, non-matured; PM, partial maturation; IVH, intraventricular haemorrhage; PVL, periventricular leukomalacia; NEC, necrotising enterocolitis; ROP, retinopathy of prematurity; PDA, patent ductus arteriosus; BPD, bronchopulmonary dysplasia; MV, mechanical ventilation.

**Table 5 jcm-11-00020-t005:** Comparison of the non-matured population by place of birth.

	NM_T (*n* = 22)	NM_S (*n* = 19)	Significance (*p*) *	OR (95%CI) Unadjusted	OR (95%CI) Adjusted ^‡^
Exitus	5 (22.7%)	1 (5.3%)	0.11	0.18 (0.02–1.78)	
Serious outcome	10 (45.5%)	12 (62.2%)	0.25	2.05 (0.58–7.21)	
Severe IVH	3 (13.6%)	4 (21.1%)	0.68	1.68 (0.32–8.73)	
PVL	2 (9.1%)	3 (15.8%)	0.64	1.87 (0.27–12.61)	
NEC ≥ 2nd degree	1 (5%)	0	1	0.95 (0.85–1.05)	
Treated ROP	2 (9.1%)	3 (15.8%)	0.64	1.87 (0.27–12.61)	
Arterial hypotension	8 (38.1%)	9 (47.4%)	0.55	1.46 (0.41–5.15)	
Treated PDA	6 (27.6%)	8 (42.1%)	0.31	1.93 (0.52–7.17)	
BPD	3 (17.6%)	7 (38.9%)	0.16	2.9 (0.62–14.22)	
Surfactant requirement	18 (81.8%)	15 (78.9%)	0.81	0.83 (0.17–3.91)	
MV	11 (50%)	17 (89.5%)	0.007	8.5 (1.57–45.91)	9.05 (1.62–50.62)
Early sepsis	1 (5%)	1 (5.3%)	0.97	1.05 (0.6–18.17)	
Late sepsis	5 (25%)	9 (47.4%)	0.14	2.7 (0.7–10.46)	
			Significance (*p*) ^†^		
MV time (hours)	132.18 ± 186.91	177.05 ± 172.92	0.52		
Maximum FiO_2_	45.8 ± 27.96	55.16 ± 24.17	0.27		

Variables are expressed as *n* (%) and mean ± standard deviation for qualitative and quantitative variables, respectively. * Fisher’s exact test/chi-squared test; ^†^ Student’s *t*-test. ^‡^ Results of multivariate analysis are shown for significant variables in the unadjusted model. NM_T, not-matured (tertiary hospital); NM_S, not-matured (secondary hospital); IVH, intraventricular haemorrhage; PVL, periventricular leukomalacia; NEC, necrotising enterocolitis; ROP, retinopathy of prematurity; PDA, patent ductus arteriosus; BPD, bronchopulmonary dysplasia; MV, mechanical ventilation.

**Table 6 jcm-11-00020-t006:** Comparison of the population with partial maturation by place of birth.

	PM_T (*n* = 28)	PM_S (*n* = 7)	Significance (*p*) *	OR (95%CI)
Exitus	6 (21.4%)	1 (14.3%)	1	0.61 (0.06–6.10)
Serious outcome	9 (32.1%)	2 (28.5%)	1	0.8 (0.13–5.22)
Severe IVH	4 (14.2%)	1 (14.2%)	1	1 (0.09–10.66)
PVL	0	0	-	-
NEC ≥ 2nd grade	0	0	-	-
Treated ROP	0	0	-	-
Arterial hypotension	9 (32.1%)	2 (28.5%)	1	0.8 (0.13–5.22)
Treated PDA	13 (46.3%)	3 (42.8%)	0.94	0.8 (0.16–4.6)
Moderate/severe BPD	3 (13.6%)	0	1	0.8 (0.73–1.02)
Surfactant requirement	22 (78.5%)	6 (85.7%)	0.64	1.6 (0.16–16.3)
MV	16 (57.1%)	3 (42.8%)	0.67	0.5 (0.1–2.9)
Early sepsis	1 (3.6%)	0	1	0.9 (0.89–1.03)
Late sepsis	9 (32.1%)	1 (14.3%)	0.64	0.3 (0.03–3.37)
			Significance (*p*) ^†^	Adjusted significance (*p*) ^‡^
MV time (hours)	113.75 ± 79.82	19.33 ± 16.04	<0.01	0.46
Maximum FiO_2_	45.92 ± 28.18	28.42 ± 11.04	0.09	0.12

Variables are expressed as *n* (%) and mean ± standard deviation for qualitative and quantitative variables, respectively. * Fisher’s exact test/chi-squared test; ^†^ Student’s *t*-test. ^‡^ Results of multivariate analysis are shown for significant variables in the unadjusted model. PM_T, partial maturation (tertiary hospital); PM_S, partial maturation (secondary hospital); IVH, intraventricular haemorrhage; PVL, periventricular leukomalacia; NEC, necrotising enterocolitis; ROP, retinopathy of prematurity; PDA, patent ductus arteriosus; BPD, bronchopulmonary dysplasia; MV, mechanical ventilation.

## Data Availability

The data presented in this study are available on request from the corresponding author. The data are not publicly available due to data protection policies.

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
