# Peer review of "Benefits of a Single Dose of Betamethasone in Imminent Preterm Labour"

_jcm, 2021, doi:10.3390/jcm11010020_

Round 1

Reviewer 1 Report

Thank you for your work. I found the manuscript quite interesting, providing useful insight on a practical, everyday issue in obstetrics and neonatology. As a general comment, I would strongly suggest that you run this past a native or fluent English speaker to further address language issues. 

Author Response

Thank you very much for your comments.
As you suggested, we have requested the revision by the mdpi translation service, to improve the English of the article.
We have added a comment in the introduction and in the discussion to emphasize that the minimum time interval in which prenatal corticosteroids begin to take effect on lung maturation has not been exactly proven.
Our study reveals that with less than 3 hours from the administration of a dose of betamethasone to delivery, better outcomes are obtained in the preterm population.
Regarding the methods, a positive blood culture was necessary to qualify as sepsis.
We hope it continues to be an article of interest to you.
Sincerely

Reviewer 2 Report

All good. Important clinical issue, but I don't see any new message from this paper. Even in the perspective of existing in references data. Study groups are relatively small, language is rather 'spanglish' than english. In perspective of present recommendations as well cochrane metanalysis novelty of the study is low. Bay the way, I cannot imagine that we may have different drug recommendations depending on level of NICU... 

Author Response

Thank you very much for your comments.
As you suggested, we have requested the revision by the mdpi translation service, to improve the English of the article.
We have added a comment in the introduction and in the discussion to emphasize that the minimum time interval in which prenatal corticosteroids begin to take effect on lung maturation has not been exactly established.
Our study reveals that with less than 3 hours from the administration of a betamethasone dose to delivery (the mean time was 60 minutes), better outcomes are obtained in the preterm population.
With our research we intend to highlight the urgency in the administration of prenatal corticosteroid in cases of preterm labour, even though it is expected that labour will end in a short period of time (less than 3 hours).
In these situations, the corticosteroid may not be administered because it is thought that it will not give time to its beneficial effect, given the urgency of delivery.
In our study, the immature preterm population could have received at least one dose of betamethasone, since the time periods between the arrival of the pregnant woman to the hospital and the birth of the premature newborn were similar.
The importance lies in the fact that although our population is not very large, the differences are appreciable. The decision to administer the corticosteroid can change the prognosis of the premature newborn.
The comparison according to the NICU level aims to highlight the importance of caring for the preterm population in tertiary hospitals. Probably a larger sample size would be needed to find significant differences between the population of preterm infants born in a level III NICU, however the trend observed is towards better results for the population that receives at least one dose compared to the immature ones.
We hope it continues to be an article of interest to you.
Sincerely

Round 2

Reviewer 2 Report

Ok.

I understand your point of view. I appreciate your explanation. Significant language improvement is easily visible, what makes your paper readable. If you fight for the paper there are some flaws to be solved.

If you present perspective of potential positive outcomes of just one dose of betamethasone I would suggest to add in your manuscript; even in title word "single" -> "single dose of... " It would be beneficial for the readers in many spots of your paper. 

Moreover, in the tables you need to use dots instead of comas.

In figure 1 should be "PM" instead of "MP"

In table 1 verify what is a proper value of CM group 148 or 152 because % are wrong and as I understood 152 should be there... 

There were no CM cases in the Secondary Hospital. If yes, analysis for 3 study subgroups shouldn't be presented in the study in which you want to compare results in between II and III level hospitals... You should then include NM and PM only... What is the rationale for the CM inclusion in the analysis? 228 preterms (152 CM, not incuded in analysis between II and III hosp.) -> actually: 76 NM + PM in II and III (mostly) Hosp... 

Moreover. Why there are no analyses between NM vs. PM (vs. CM) in the II Hosp.? Due to the number of cases 19 vs. 7. I would be really careful with such strong conclusions you wrote... 

No particular types of tests are pointed in the table/s in regard to the received p-values. 

There are no pointed study limitations. 

I would decrease strength of your conclusions: "may" instead of "could" and would be careful with usage "recommendations" in regard to such study groups...

Acknowledgements should be in your case omitted as there are no particular ones...

Author Response

Thank you very much for your comments.
First of all we have accepted your suggestion to add "single" dose of betamethasone to highlight this condition.
We have changed the commas to the periods in the tables and in the text.
We have changed in figure 1, MP by PM.d
As suggested, we have withdrawn the full maturation cohort from the study. Our intention was to compare with the gold standard, but it is true that we do not have cases of preterm newborns with the complete pattern born in secondary hospitals. Because of this, we have removed it from the study.
In table 4 you can see the comparison between NM vs. P.M. indeed the sample is scarce. In secondary hospitals, there is a lower percentage of betamethasone administration in imminent delivery. For this reason, our research tries to highlight the benefits of its administration and encourage the initiation of lung maturation even if delivery occurs in a short period of time.
We have added the tests used to the p-values ​​of the tables.
We have expanded the limitations of the study.
We have reduced the strength of the conclusions as suggested.
We have withdrawn the acknowledgments.

Thank you very much again for your comments.
We hope that our investigation is of interest to you.
